# Cerebral Microbleeds Associate with Brain Endothelial Cell Activation-Dysfunction and Blood–Brain Barrier Dysfunction/Disruption with Increased Risk of Hemorrhagic and Ischemic Stroke

**DOI:** 10.3390/biomedicines12071463

**Published:** 2024-07-01

**Authors:** Melvin R. Hayden

**Affiliations:** Department of Internal Medicine, Endocrinology Diabetes and Metabolism, Diabetes and Cardiovascular Disease Center, University of Missouri School of Medicine, One Hospital Drive, Columbia, MO 65211, USA; mrh29pete@gmail.com

**Keywords:** Alzheimer’s disease, blood–brain barrier, cerebral amyloid angiopathy, cerebral microbleeds, dementia, hypertension, magnetic resonance imaging, neurovascular unit, cerebral small vessel disease, transmission electron microscopy

## Abstract

Globally, cerebral microbleeds (CMBs) are increasingly being viewed not only as a marker for cerebral small vessel disease (SVD) but also as having an increased risk for the development of stroke (hemorrhagic/ischemic) and aging-related dementia. Recently, brain endothelial cell activation and dysfunction and blood–brain barrier dysfunction and/or disruption have been shown to be associated with SVD, enlarged perivascular spaces, and the development and evolution of CMBs. CMBs are a known disorder of cerebral microvessels that are visualized as 3–5 mm, smooth, round, or oval, and hypointense (black) lesions seen only on T2*-weighted gradient recall echo or susceptibility-weighted sequences MRI images. CMBs are known to occur with high prevalence in community-dwelling older individuals. Since our current global population is the oldest recorded in history and is only expected to continue to grow, we can expect the healthcare burdens associated with CMBs to also grow. Increased numbers (≥10) of CMBs should raise a red flag regarding the increased risk of large symptomatic neurologic intracerebral hemorrhages. Importantly, CMBs are also currently regarded as markers of diffuse vascular and neurodegenerative brain damage. Herein author highlights that it is essential to learn as much as we can about CMB development, evolution, and their relation to impaired cognition, dementia, and the exacerbation of neurodegeneration.

## 1. Introduction

Neuroimaging by magnetic resonance imaging (MRI) of cerebral microbleeds (CMBs) is being increasingly identified as important structural remodeling changes as hypodense small, smooth, and rounded structures (2–5 mm and sometimes up to 10 mm) on T2*-weighted gradient recall echo (T2*-GRE) or susceptibility-weighted sequences MRI images (Figure 1) [1,2,3,4].

CMBs may now be considered markers for small vessel disease (SVD) [1] that strongly suggest an increased risk for stroke (hemorrhagic and/or ischemic) and stroke mortality [3]. They are also currently associated with the clinical presence of hypertension (HTN), cerebral amyloid angiopathy (CAA), advancing age, cerebro-cardiovascular disease, SVD, late-onset or sporadic Alzheimer’s disease (LOAD), cardiovascular disease (CVD), and chronic kidney disease [1,5,6]. Notably, the Rotterdam study revealed that within the general population, those with an increase in the number of microbleeds, as demonstrated by MRI, were associated with an increased risk of stroke (both hemorrhagic and ischemic) [7]. The risk differs for the subtypes of stroke depending on the location of the cerebral microbleeds (cortical-lobar CAA-related; subcortical—deep, white matter, basal ganglia (BG), thalamus, cerebellar are (HTN)-related)) [8]. Notably, those individuals with the largest microbleed burden were at the highest risk of stroke, and microbleeds not only mark the progression of cerebrovascular pathology but also represent a precursor of stroke [7]. 

SVD may be defined as the sum of all neuropathological processes that affect small vessels of the brain, including small arteries, arterioles, capillaries, venules, and small veins [9,10,11]. Importantly, SVD consists of multiple MRI-identifiable, aberrant findings, which include recent small subcortical infarcts, lacunes, enlarged perivascular spaces (EPVS), white matter hyperintensities (WMH), CMBs, and brain atrophy of cortical neurons (Figure 2) [1,12,13].

As their name implies, CMBs are currently known to develop as a result of a small accumulation of cerebral microvessel blood components (erythrocytes—red blood cell(s) (RBCs), plasma, hemoglobin, hemosiderin or hemosiderin from surrounding macrophages) that have escaped microvessels lumens due to brain endothelial cells activation/dysfunction (BEC*act/dys)* and subsequent or concurrent blood–brain barrier dysfunction or disruption (BBB***dd***) with increased microvessel permeability and leakage of RBCs, leukocytes, fluids, and solutes. Additionally, there may be a loss of the arterial vessel walls media vascular smooth muscle cell (VSMC) integrity (due to deposition of amyloid beta in CAA and hyalinosis and/or arteriolosclerosis in hypertensive vasculopathy with rupture), which allows for the escape of luminal blood and its contents to aberrantly reside within the CNS parenchymal interstitium [1,14,15]. The escaped blood from the lumen now aberrantly residing within the neuronal interstitial parenchyma is capable of instigating a brain injury with a known response to injury wound healing mechanism. This response to injury is due to the neurotoxic contents of red blood cell (RBCs) hemoglobin metabolism, including hemosiderin, which gives these structures a flare component as well as their hypodense, black appearance on T2*-GRE MRI. Additionally, the plasma that is extruded from these microbleeds contains neurotoxic thrombin, plasmin, and the components within the complement cascade [1,15,16]. The information regarding the possibility of the BEC*act/dys* and subsequent or concurrent NVU BBB***dd*** is currently gaining a great deal of support [17,18,19,20,21]. 

During the past two decades, there has been increasing and widespread clinical and research use of MRI (~1966–2000), and our understanding of CMBs has undergone exponential growth. CMBs were originally thought to be asymptomatic markers of SVD; however, emerging data have shown an association between microbleeds and cognitive impairment—dementia, with an associated increased risk for ischemic and large symptomatic strokes [3,7,22]. CMBs are considered to be MRI-defined lesions corresponding to small deposits of blood components (due mainly to hemosiderin that accumulates in perivascular macrophages to allow MRI identification by GRE from previous episodes of small amounts of bleeding in the brain primarily into the neuronal interstitial spaces or interstitium) that are most commonly related and thought to be markers for small vessel damage, small vessel disease and concurrently raise a red flag for the increased risk of large symptomatic neurologic intracerebral hemorrhages [ICH] [1,3,7,22]. Indeed, CMBs have generated great interest as MRI GRE markers for SVD prone to bleeding, with accumulating evidence that they are related to an increased risk of stroke (hemorrhagic and/or ischemic infarcts) and thus identify those individuals who are at greater risk for the development of larger symptomatic ICH [1,3,7,22,23,24]. Therefore, it is essential that we try to learn as much as we can about their evolutionary development and how and why they result in neuronal damage in addition to ischemic and/or hemorrhagic stroke. 

In summary, knowledge has expanded greatly regarding CMBs during these past two decades [3], and they are considered to be MRI-defined lesions corresponding to small deposits of blood components (due mainly to paramagnetic qualities of hemosiderin that accumulate in perivascular macrophages to allow MRI identification by T2*-GRE from previous episodes of small amounts of bleeding in the brain primarily in the neuronal interstitial spaces or interstitium) [3]. These bleeds are thought to be markers for small vessel damage and concurrently raise a red flag for the increased risk of future large symptomatic ICH [1,3,22]. The presence of increased CMBs is related to an increased risk of stroke (ischemic and/or hemorrhagic infarcts) and thus identifies those individuals who are at greater risk for the development of larger symptomatic ICH [1,3,22,23,24]. CMBs may also predict cerebral bleeding after a stroke [25], and their presence is regarded as a marker of diffuse vascular and neurodegenerative brain damage [26]. Additionally, CMBs may be considered to be amongst the most prevalent known neurologic processes and are also known to have major implications in regard to stroke, dementia and/or cognitive impairment, and aging [9,27,28]. Therefore, it is essential that we continue to expand our database of knowledge and learn as much as we can about CMBs’ important evolutionary role in development as to how and why they may evolve. 

The primary objective of this narrative review is to provide the reader with an increased database of knowledge and understanding of CMBs, their evolution via BEC*act/dys*, and BBB*dd* with increased permeability and increased risk of hemorrhagic and ischemic stroke that are capable of exacerbating neurodegeneration. 

## 2. A Possible Sequence of Events in the Development of Cerebral Microvessel Bleeds (CMBs)

Cerebral microvessels, including small arteries, precapillary arterioles, true capillaries, postcapillary venules, and veins, are capable of becoming leaky due to increased permeability and multifactorial rupturing as a result of BEC*act/dys* and/or BBB***dd*** [29,30]. 

### 2.1. Brain Endothelial Cell Activation and Dysfunction (BECact/dys) 

The monolayer of BECs, similar to systemic ECs, plays an important multifunctional role in cerebrovascular homeostasis by regulating blood fluidity, fibrinolysis, vascular tone, permeability, angiogenesis, leukocyte, RBC, and platelet adhesion as well as aggregation [31]. Importantly, BECs may also be considered as the gate keepers and sentinel cells between the circulating blood and neuronal parenchyma and in their quiescent state (non-activated) in order to protect the brain from peripheral neurotoxins in health along with its NVU BBB, pericytes, ECM and tightly adherent perivascular astrocytes. BEC*act/dys* indicates activation of the cellular machinery that upregulates cell-surface inflammatory adhesion proteins such as vascular cellular adhesion molecule-1 (VCAM-1), intercellular cellular adhesion molecule-1 (ICAM-1), and endothelial leukocyte adhesion molecule (ELAM or E-selectin) in order to call up peripheral leukocytes for their adherence to the endothelium (activation). 

I-CAM, V-CAM, and E-Selectin become activated by peripherally-derived systemic injurious stimuli, including ***p***CC, such that there is an impaired synthesis of NO by the endothelial nitric oxide synthase (eNOS) enzyme. This may occur concurrent with eNOS uncoupling due to oxidation of tetrahydrobiopterin (BH4) oxidation to BH3 or BH2. BH4 must be completely reduced in order to run the eNOS reaction to generate NO to result in decreased bioavailable NO [17,32,33,34]. Further, BECs may also be considered as gate keepers and sentinel cells between the circulating blood and neuronal parenchyma and the BEC in its quiescent state (non-activated) to protect the brain from peripheral neurotoxins in health along with its NVU BBB, pericytes, ECM and tightly adherent perivascular astrocytes. 

There are multiple injurious species from the peripheral circulation, which are capable of instigating BEC*act/dys* with upregulation of inflammatory signaling as well as promoting a decrease in bioavailable nitric oxide since these two frequently present concomitantly (Figure 3) [13,16,35]. 

Once the NVU undergoes BECact/dys, the BEC is capable of synthesizing and secreting numerous brain-derived cytokines and chemokines specifically from activated BEC (***bec***CC or ***cns***CC) to contrast with ***p***CC. These ***bec***CCs include the following cytokines: Interleukin-1beta (IL-1β), interleukin-6 (Il-6), interleukin-8 (IL-8), and tumor necrosis alpha (TNFα), while chemokines include (MCP-1) or (CCL2), (CCL5) or (RANTES), plus others that, in turn, stimulate the brains’ reactive microglia and astrocytes (rMGCs, rACs) to produce even more ***cns***C/C. Additionally, the BECs, rMGCs, and rACs are also capable of actively secreting reactive oxygen species (ROS), resulting in the creation of the reactive species interactome (RSI) of reactive oxygen, nitrogen, sulfur species (RONSS), which in turn activate local matrix metalloproteinases -2, -9 (constitutive MMP-2 and inducible MMP-9, respectively) that are capable of degrading the glia limitans allowing the proinflammatory leukocytes to breech this second outermost barrier of the perivascular space to result in neuroinflammation (Figure 4 and Figure 5) [15,36,37].

Importantly, Step 2 in Figure 4 also allows for the free extravasation of blood components, including neurotoxic red blood cells (hemoglobin metabolic byproducts including hemosiderin, which allows for the hypodensities on GRE MRI images and plasma that contains neurotoxic thrombin, fibrin, plasmin, hemoglobin metabolic byproducts such as hemosiderin and free iron to instigate further neuroinflammation, oxidative stress and activation of local MMPs that contribute to ongoing degradation of the PVS outer barrier (glia limitans) in a vicious cycle once it is instigated). Note that the dysfunctional pvACef AQP4 water channel is associated with the dysfunctional bidirectional signaling between the neurons (N) and the dysfunctional pvACef (Figure 4 and Figure 5). 

In either a step-wise fashion or concurrently, BEC*act/dys* gives rise to BBB dysfunction or disruption [17]. 

### 2.2. Blood–Brain Barrier Dysfunction and/or Disruption (BBBdd) with Increased Permeability

Wang et al. make a strong statement in that they state the following: “Blood–brain barrier (BBB) dysfunction or disruption (BBB***dd***) is considered to be the event that initiates CMBs development.” [17]. This group believes strongly that BBB***dd*** is responsible for CMBs, as put forth earlier in the text, along with multiple other authors [18,19,20,21]. Risk factors for the development of CMBs consist of advancing age, HTN, CAA, type 2 diabetes mellitus (T2DM), smoking, and previous strokes [1,3,17]. 

Once BEC*act/dys* has occurred, this allows for the development of BBB***dd*** due largely to the activation of redox-sensitive MMPs via inflammation generated by cytokine ROS production. These MMPs not only promote the dysfunction and degradation of the TJ/AJs but also allow for the degradation of the glia limitans, the outermost barrier of the PVU, to allow for Step 2, as portrayed in Figure 5, to allow the escape of leukocytes, red blood cells, and plasma into the surrounding parenchymal interstitial spaces to result in neuroinflammation and CMBs as in Figure 4 and Figure 5 [36]. Thus, BEC*act/dys* may be considered a promoter of BBB***dd***, which in turn allows for the breeching of the PVU/PVS glia limitans the outermost barrier of the perivascular space to enter the neuropil interstitial spaces (Figure 4 and Figure 5). 

Wang et al. conclude with the following paragraph as follows; “In conclusion, despite many details that still require study, considerable evidence suggests that BBB dysfunction appears to play a significant role in the development and progression of CMBs”. Risk factors for CMBs can exacerbate BBB breakdown through the vulnerability of the BBB to anatomical remodeling and functional changes [17]. Indeed, BBB***dd*** allows for increased permeability, and this could lead to the extravasation of RBCs from cerebral microvessels into the neuropil interstitial space, which leads to the deterioration of the brain’s environment and further aggravate brain degeneration due to the development of CMBs [17,18,19,20,21,38]. Additionally, the next or concurrent steps in the progression to CMBs would be one of two primary etiologies or mechanisms, such as HTN or CAA, to develop CMBs. Notably, primarily basal, deep, infratentorial, or occasionally mixed CMBs develop in hypertensive vasculopathy, and lobar or cortical CMBs develop almost exclusively in CAA [2,7]. 

Cerebral microvessels, including small arteries, precapillary arterioles, true capillaries, postcapillary venules, and veins, are capable of multifactorial extravasation or rupturing, and these are a known primary source for CMBs that involve the vicious cycle of oxidative stress and inflammation to trigger the development of the neurodegenerative cascade and impaired cognition (Figure 6) [1,17,29,30]. 

### 2.3. Hypertensive (HTN) Vasculopathy

HTN is the second leading cause of CMBs following advancing age [17] and is a major risk factor for CMBs [29,39]. CMB pathogenesis is thought to be primarily a result of the vicious cycle of oxidative stress and inflammation, as in Figure 6, and their remodeling effects on the VSMCs, which results in a loss of integrity and allows these cerebral microvessels to be prone to rupture [29]. Additionally, these hypertension-induced microbleeds are also worsened by aging [40] and amyloid pathology [41]. Cerebral microbleeds are associated with worse cognitive function, and underlying mechanisms may involve not only local brain injury but also chronic inflammation [42]. HTN-related CMBs are known to occur primarily in the subcortical—deep, white matter, BG, thalamus, brain stem, and cerebellar regions [3,8].

HTN affects each of the cell-types in the NVU by various mechanisms, which result in BECact/dys, BBB***dd***, and impaired neurovascular unit (NVU) function with impaired neurovascular responsiveness. Endothelial dysfunction results from reduced NO bioavailability as a result of impaired eNOS function via reduced expression, mislocalization, impaired phosphorylation, and eNOS uncoupling, thus resulting in reduced NO production and decreased bioavailability and increased ROS production by perivascular macrophage(s) (PVMΦs) [43]. Subsequently, HTN allows cerebral microvessels to become a net producer of damaging ROS instead of vasculoprotective NO. 

Angiotensin II (AngII)) type 1 receptor (AT1R) activation results in the activation of NADPH oxidase 2 in both BECs and PVMΦs and a potential interaction with toll-like receptor 4 (TLR4) in ECs, which contribute to dysfunction and disruption of the BBB or BBB***dd***. Neurovascular coupling is impaired by PVMΦs-derived ROS, aldosterone-induced damage of inwardly rectifying potassium channel 2.1 (K_IR_2.1), and endothelial hyperpolarization, as well as altered calcium signaling in astrocytic endfeet. Also, pericytes express Nox4 (NADPH oxidase 4), which is upregulated by AngII and may contribute to vascular inflammation [43,44]. 

HTN is associated with BEC*act/dys* and BBB***dd***, and once these are established, they play an important role in the further development of CMBs and their progression with increased permeability and leakage of RBCs [17,43,44]. Importantly, inflammation, pCC, becCC, cnsCC, ROS, SNS, RAAS, and specifically Ang II are now known to play important roles in the development and perpetuation of HTN [17,43,44]. 

### 2.4. Cerebral Amyloid Angiopathy (CAA) 

CAA is a leading cause of cognitive impairment and ICH in the elderly global population [45,46]. Indeed, CAA may be found to be present in ≥50% of individuals over the age of 80 [47,48]. Of course, some will also have diagnostic changes compatible with LOAD-like dementia and VaD. This represents a red flag since, in three more years, the baby boom generation will begin to turn 80 at a rate of 1000 or more a day over the next 10.5 years. Thus, CAA will be co-occurring with LOAD and T2DM. 

CAA may be characterized by the deposition of primarily misfolded amyloid beta (Aβ (1–40)) proteins in the media and adventitia of small and mid-sized cerebral arteries (less commonly in the capillaries and veins) and leptomeninges [47]. It occurs most often in its sporadic form; however, mutant variants are known and frequently familial. Its clinical presentation includes strokes, both ischemic and hemorrhagic presentation, with primary intracerebral hemorrhage being most common due to ruptured vessels with bleeding. Aβ (1–40) is known to be vasculotoxic to the media’s VSMCs and results in a loss of integrity and weakness of the arterial media with subsequent rupture. Aβ (1–40), which is a product of amyloid precursor protein (APP) cleaved by β-secretase 1 (BACE-1) and γ-secretases, is the primary protein deposited in these microvessels [48,49]. 

Aβ aggregation and deposition due to either excess production or impaired clearance by perivascular spaces are known to be vasculotoxic to both BECs and VSMCs, resulting in a loss of integrity of the microvessel media, which predisposes these arterioles to rupture, allowing the formation of CMBs. For example, in sporadic CAA and cerebral autosomal dominant arteriopathy with subcortical infarcts and leukoencephalopathy (CADASIL), accumulation of misfolded Aβ protein deposits without evidence of increased production strongly suggests impairments in clearance [50]. As Aβ (1–40) continues to deposit over time in the media [51], it results in the loss of vascular integrity, and ruptures subsequently develop with the extravasation of fluids, solutes, and RBCs that form CMBs, which also activate perivascular macrophages and associates with inflammation. These CMBs reside in the lobar or cortical regions in contrast to the deep infratentorial basal ganglia hypertension-derived CMBs [8]. Additionally, the extruded Aβ and inflammation will have an effect on the BECs to result in BEC*act/dys* that may also instigate BBB***dd*** with increased permeability to result in further CAA-associated injury [52]. 

Also, Aβ is capable of disrupting the BEC mitochondrial metabolic pathways by inhibiting the tricarboxylic acid cycle, electron transport chain, and oxidative phosphorylation [53,54,55]. 

Interestingly, we have previously treated streptozotocin-induced diabetic mice that developed BBB dysfunction and disruption thought to be due to the glucotoxicity effect on BEC mitochondria and excess mitochondrial ROS production. Topiramate is a mitochondrial-specific carbonic anhydrase inhibitor used clinically as an antiseizure medication which serves as specific mitochondria carbonic anhydrase inhibitor (antioxidant), and topiramate treatment prevented BBB dysfunction and disruption increased permeability as measured by ^14^C-sucrose measurements as well as protecting tight and adherent junction BBBs from attenuation and loss of tight and adherens junction by ultrastructure studies (Figure 7) [56]. 

Notably, Aβ (1–40) is capable of binding to the receptor for advanced glycation end-products (RAGE) to generate ROS, which is capable of activating MMP-2 and -9 that are capable of degrading tight and adherens junctions (TJ/AJ) to result in BBB***dd*** with increased permeability [57,58]. Also, Aβ (1–40) is capable of activating BEC to promote the secretion of proinflammatory ***bec***CC, including Il-1β, IL-6, and MCP-1 molecules that are capable of recruiting even more peripheral immune cells into the brain [59]. 

In summary, CAA is a definite age-related disease. For example, in the Religious Orders Study (404 individuals), 84% of participants had CAA [60], and this study also found that CAA frequently co-occurred with LOAD, with an estimated 78–98% of individuals with LOAD also having CAA [61]; however, only ~25% of patients with LOAD also have severe CAA [62]. 

CAA and HTN are the most common clinical causes of CMBs after advanced aging [63]; therefore, a table comparing CAA and HTN similarities and differences in relation to how they are associated with the development of CMBs that are identified by (2–5 mm and sometimes up to 10 mm) on T2*-GRE or susceptibility-weighted sequences MRI images are presented (Table 1). 

Since there are considerable similarities between CAA and HTN, could there be a synergism if they co-occurred? Indeed, HTN is frequently observed in CAA individuals [64], and previous studies have suggested that HTN may accelerate CMBs in CAA [65]. Further, microscopic and immunohistochemistry studies have previously shown that HTN-related arteriolosclerosis and CAA pathological changes of AB (1–40) deposition in the media VSMC regions often co-exist [66,67]. Interestingly, Zhu et al. sought to find if there was an association between HTN vasculopathy and CAA by studying MRIs of 222 individuals who presented with ICHs. They studied 222 (mean age of 59.88 ±13.56) and found a significant association between HTN vasculopathy and CAA and SVD in these individuals. They felt these findings suggested a possible synergistic effect between HTN vasculopathy. CCA and SVD in ICH; however, further studies will be required to answer this proposed question with certainty [68]. 

## 3. Transmission Electron Microscopy (TEM) Imaging of BEC*act/dys*), BBB*dd* with CMBs

CMBs may be present even before we can observe them on MRI by histopathology and ultrastructural TEM studies. Previous studies have determined that there is a link between T2DM and the development of LOAD (Alzheimer’s disease) [67] and that there are underlying mechanisms that have been proposed to help explain the association of T2DM and cognitive impairment, which include BEC*act/dys*, BBB***dd***, inflammation, and insulin resistance [69,70,71,72,73,74]. Now, according to Teng et al. recent findings, we can now add the existence of SVD, which includes CMBs that are included to underlie an additional cause of cognitive impairment along with lacunes, EPVS, and WMH [68]. Notably, these possible mechanisms are also important factors that contribute to the pathogenesis and development of SVD, which contributes to cognitive impairment [75]. 

Importantly, obesity, metabolic syndrome (MetS), and T2DM are known to be associated with significant ultrastructure TEM brain remodeling with the development of SVD, cognitive impairment and dysfunction (CID), vascular cognitive impairment and dementia (VCID), MCI and depression with SVD including EPVS and CMBs. Notably, in the 20-week-old female obese, insulin resistance, MetS, and T2DM preclinical diabetic *db/db* mouse models, CMBs were found only in diabetic *db/db* mice and not control models or those *db/db* models treated with the antidiabetic medication empagliflozin [15,76,77,78,79,80,81,82,83,84]. Previously, in the *db/db* models, we were able to make multiple observations of BEC*act/dys*, including BEC abrupt thickening and loss of cytoplasm electron density, BEC basement membrane thickening, with aberrant vacuole-like bodies, leukocyte, red blood cell, and platelet adherence to the activated BECs (Figure 8) [15,83,84].

Importantly, we were able to observe the presence of CMBs by observing the free homogeneous electron-dense regions of free plasma or RBCs that had previously extruded from adjacent small NVU capillaries. Even though these NVU-imaged capillaries appeared to be intact without observable disruption, one cannot deny the escape or extrusion of their luminal electron-dense and homogeneous plasma contents and RBCs to reside freely within the interstitial spaces of the adjacent neuropil (Figure 9, Figure 10 and Figure 11) [13,15,83,84]. 

The NVU capillary BECs in this model had intact TJ/AJ by ultrastructure observations in these regions; however, overall, they demonstrated multiple changes of BEC*act/dys* such as pericyte endfeet retraction, basement membrane thickening, perivascular astrocyte endfeet detachment and retraction [13,83]. Importantly, Wang et al. have noted that the findings of CMBs are dependent upon the paramagnetic properties of hemosiderin or erythrocytes that have passed through BEC that have developed BECact/dys and feels that it is worth considering CMBs without microvessel rupture as shown in the above obese diabetic *db/db* models (Figure 9, Figure 10 and Figure 11) to be worthy of further studies [17]. 

Notably, obesity, T2DM and MetS have been found to be associated with an increased risk and an association with lobar CMBs in T2DM [4] and deep CMBs in metabolic syndrome [79]. Importantly, in our diabetic *db/db* models, the CMBs were all located in the lobar cortical regions III; however, they were very close to the transition zones between layers III and the white matter regions. 

We have now observed and learned that CMBs manifested by plasma and RBC extrusions from adjacent small NVU capillary microvessels might occur early by 20 weeks of age in the preclinical female diabetic *db/db* mouse models (Figure 9, Figure 10 and Figure 11). Therefore, it now becomes somewhat evident that CMBs may also be part of a spectrum disorder, as are the other aberrant remodeling changes found with SVD, such as lacunes, EPVS, and WMHs [13]. Also, the previous spectrum figure describing these spectrum changes regarding EPVS may also be pertinent to the development and evolution of CMBs (Figure 12) [13].

### Oxidative—Redox Stress: Implications in the Development of CAA, LOAD, SVD, and CMBs

Oxidative—redox stress, accelerated atherosclerosis, MetS, and T2DM all have complex interactions in the development of microvessel remodeling pathology that is associated with LOAD, CAA, VaD, and SVD, which includes CMBs. There are multiple injurious stimuli resulting in the accumulation of excess ROS signaling in the CNS (Figure 13) [69].

Notably, Han et al. were able to demonstrate that ROS (oxidative-redox stress) is a critical contributor to (i) CAA formation, (ii) CAA-induced vessel dysfunction, and (iii) CAA-related microhemorrhage (CMBs). They were able to demonstrate that apocynin (a NAPDH Ox NOX2 inhibitor) significantly decreased CAA and CMBs in the Tg2576 mouse; however, the non-specific ROS scavenger tempol decreased these variables were not significant. Thus, ROS and, in particular, NADPH oxidase NOX2-derived ROS appear to be promising therapeutic targets for individuals with CAA, CMBs, and LOAD [85]. 

Since the ultrastructural TEM images in this section were obtained from the obese, MetS, T2DM diabetic *db/db* female models at 20 weeks of age, it is appropriate to discuss why these models demonstrated CMBs in the frontal cortical layers III. T2DM is known to increase the risk for sporadic LOAD [69]. The author has previously suggested that there are at least five major intersecting links to be considered with this increased risk as follows: (i) Advancing age, since both T2DM and LOAD are age-related diseases; (ii). metabolic alterations (hyperglycemia and advanced glycation end products along with its receptors (AGE/RAGE) interactions increasing ROS and hyperinsulinemia-insulin resistance that remains a linking linchpin between T2DM and LOAD); (iii) oxidative stress (RONSS—reactive species interactome); (iv) Peripheral metainflammation (***p***CC) and central neuroinflammation (***cns***CC); (v) Vascular (macrovascular accelerated atherosclerosis—vascular stiffening and microvascular NVU remodeling with impaired cerebral blood flow [70]. The vicious cycle of oxidative-redox stress (iii) and pCC/cnsCC metainflammation and neuroinflammation (iv), respectively, as in Figure 6, may indeed be the trigger for the development and evolution of increased CAA, LOAD, SVD, and CMBs with increased permeability due to BEC*act/dys* and BBB***dd***, which contributes to impaired cognition and neurodegeneration (see discussion in Section 2.2). The oxidative–redox stress hypothesis has been in existence for some time in regard to the development and progression of LOAD and synaptic dysfunction or loss and neurodegeneration with impaired cognition [86]. 

Further, Vargas-Soria et al. have demonstrated that CAA and related vascular remodeling are present in mixed murine models utilizing AD-T2DM (APP/PS1x*db/db*) by crossbreeding APPswe/PS1dE9 mice [87]. This model demonstrated that T2DM significantly affects vascular pathology and CAA deposition, which is increased in AD-T2D mice, suggesting that T2D favors vascular accumulation of Aβ. Moreover, T2D synergistically contributes to increased CAA-mediated oxidative stress and MMP activation. Importantly, this group concluded that even the early crosstalk between metabolic disease (found in prediabetes, obesity, and insulin-resistant models induced by a high-fat diet) could contribute to the increased interstitial plaque or vascular-related CAA that affects vascular integrity and contributes to AD pathology and associated aberrant functional changes in the brain microvasculature. 

In summary, oxidative-redox stress plays a critical role in the pathogenesis of CAA, CMBs, HTN, LOAD, and VaD with impaired cognition, synaptic dysfunction and or loss, and neurodegeneration via numerous mechanisms, including BEC act/dys, BBB***dd*** with increased permeability, mitochondria dysfunction with increased mitochondrial ROS, ROS-induced cnsCC neuroinflammation, MMP activation, as well as arteriole VSMC dysfunction with leakage and VSMC apoptosis with loss of the integrity of media allowing for the formation of rupture-prone arterioles and leakage of blood contents into the interstitial spaces wherein the iron content will accelerate the formation of even more ROS and redox stress via the Fenton reaction to damage neurons and increase the production of soluble Aβ and oligomeric forms that deposit not only in the vascular walls but also increase Aβ plaque formation within the interstitial space, which in turn can result in neurofibrillary tangles and tau formation [51,88,89,90,91,92,93]. 

## 4. Atrial Fibrillation (AF) Association with Cerebral Microbleeds and Stroke

AF is the most common cause of cardioembolism and stroke, and long-term oral anticoagulation is the mainstay for therapy [94]. Also, it is known that stroke (hemorrhagic and ischemic) risk is increased by three to five-fold in those individuals with chronic AF [95]. Unfortunately, studies suggest that individuals in general with CMBs, both with and without AF, are at an elevated risk for future stroke and, in particular, ICH [24,96,97]. Importantly, the prevalence of CMBs is significantly higher in individuals with chronic AF when compared to those without AF [98]. 

AF is one of the most common arrhythmias and is known to increase with aging, similar to CMBs, and its prevalence is increasing within our global aging population [99]. Further, AF results in cerebrovascular dysfunction with impaired cerebral blood flow (CBF) [100]. Additionally, Junejo et al. have provided solid evidence for diminished cerebral blood flow, cerebral autoregulation, neurovascular coupling, and cerebrovascular carbon dioxide reactivity, which supports diminished cerebrovascular vasodilatory reserve in AF patients when compared to control participants in sinus rhythm [100]. It was recently found (March 2023) that in those individuals who have AF and are on antithrombotic therapy for secondary prevention after ischemic stroke or transient ischemic attack, the presence of CMBs was associated with increased risk of both subsequent intracerebral hemorrhage and ischemic stroke with a greater association for increased intracerebral hemorrhage [101]. 

An algorithm has been recently proposed by Fisher. which incorporates MRI screening into the anticoagulation decision-making protocols [102]. Their algorithm includes individuals with chronic non-valvular AF based on ≥age 60 who should have MRI screening prior to beginning oral anticoagulation. Fisher has recommended starting oral anticoagulation in those who have none or less than five subcortical CMBs. However, in those with any or ≥five CMBs, he recommends neurologic consultation and avoiding warfarin in preference to using non-oral anticoagulants if, indeed, anticoagulation is still thought to outweigh the possible side effects of ICH [102]. Also, he recommends repeating MRIs for comparison who develop neurological deficits and discontinuing anticoagulation if there is a progression of CMBs. While this approach may not be accepted by all, it does importantly incorporate the detrimental role of increased CMBs and the role of personalized medicine [102]. 

Stroke (increased up to five-fold) is the leading complication of chronic non-valvular AF [95,103], and 80% of strokes are caused by arterial occlusion of cerebral arteries, whereas the remaining 20% are caused by intracerebral hemorrhages [104]. Unfortunately, this presents somewhat of a conundrum in that antithrombotic therapy can act like a double-edged sword with both beneficial and adverse effects, and, therefore, the benefit–risk ratio must be considered at all points in therapy [105]. Yet, oral antithrombotic therapy remains the best treatment option to prevent cardioembolism in AF [106]. Because of the above associations and the everchanging landscape regarding anticoagulation in association with CMBs, it is recommended that the most up-to-date recommendations regarding anticoagulation be followed. 

## 5. Apolipoprotein E Association with CMBs

Apolipoprotein E is known to be important in lipid metabolism, lipid transport, and membrane biosynthesis in sprouting and synaptic remodeling [107]. The presence of the APOE-ε4 allele increases the amount of Aβ accumulation in the brain (vascular and parenchymal) and when the amount ofAPOE-ε4 was increased, this also paralleled the prevalence of CMBs [108]. Ingala et al. also found that CMBs occurred in lobar regions and co-localized with white matter hyperintensities. Interestingly, they found that the APOE-ε2 allele did not protect from developing CMBs, whereas the allele APOE-ε3 was neuroprotective [108]. Additionally, this group found that those individuals homozygous for the APOE-ε4 genotype had more fragile microvessels in lobar locations and co-occurred with WMH, which suggested increased vascular vulnerability for the development of CMBs [108]. 

## 6. Conclusions

This narrative review suggests that BEC*act/dys* and subsequent or concurrent BBB***dd*** play a key and central role in the pathophysiology of CMBs and their increased risk for ICH as well as cerebral SVD. However, their occurrence and increased risk for ICH may still need further confirmation in larger longitudinal studies in human individuals.

Narrative reviews provide a flexible yet rigorous platform to approach a topic of interest and present information organization and synthesis that is extremely useful for sharing knowledge and information with other educators and researchers. However, it is important to note that they also have inherent limitations in that the author(s) biases may readily creep into the review. 

The finding of CMBs in numbers ≥ 10 is indeed a “red flag” for clinicians and **re**searchers and has become an important and independent predictor for the increased risk of intracerebral hemorrhage (ICH) [74]. Further, in the author’s opinion, this risk of ICH is strengthened if the CMBs occur in mixed regions of the brain, i.e., both lobar and deep—infratentorial regions that may indicate the presence of both CAA and HTN vasculopathy and a recent publication pointed to the importance of the role of simultaneous multiple intracerebral hemorrhages that are characterized by symptomatic intracerebral hemorrhages within different arterial territories and some had both CAA and hypertensive vasculopathy. While this study did not answer the author’s opinion regarding synergism, it nevertheless pointed to the co-occurrence and possible importance of CMBs occurring simultaneously in lobar and deep (BG) regions. [109]. 

The presence of CMBs in numbers ≥ 10 suggests that there is a significant degree of microvessel disease with increased SVD presence with loss of microvessel integrity and increased vulnerability to undergo extravasation of microvessel blood luminal contents to undergo diapedesis and/or rhexis (rupture). The subsequent increase in CMBs may increase the risk for both ischemic and hemorrhagic intracerebral stroke. However, based on the recent literature and current evidence, the presence of CMBs should not be a contraindication to intravenous thrombolysis for the treatment of acute cerebral infarction [110]. Additionally, Chacon-Portillo et al. have stated that CMBs should not dictate the treatment of acute stroke [111]. 

There are considerable studies that suggest that aging, APOE-ε4, HTN, peripheral systemic inflammation, and neuroinflammation correlate with the increased risk for the development of CMBs, as well as the discussion in Section 2 of this text (Figure 14) [17,42,112,113,114,115]. 

Currently, it is becoming more and more obvious, considering the clinical significance of CMBs, that it is necessary to place a greater emphasis on studying the development and progression of CMBs. Many details regarding the development and evolution of CMBs still require further study, and even though there is considerable evidence that BEC*act/dys* and BBB***dd*** appear to play a very important and key role in the development in the development of CMBs and stroke (both ischemic and intracerebral hemorrhage), more studies are necessary.

The author suggests that prior to examining the present and future directions, it is important to examine the past from a historical perspective. Not only does the past allow us to study what has been, but it allows us to get to know some of the giant shoulders upon which we now stand today and will continue to so far into the future (Figure 15).

Herein, the author presents a brief and incomplete list of influential authors (Wardlaw J.M. et al.; Greenburg S.M. et al.; Offenbacher H. et al.; Cordonnier C. et al.; Shoamanesh A. et al.; Vernooij M.W. et al.; Veluw S.J. et al.; Ghaznawi R. et al.; Smith E.E.) and references that reflect the past work and progress of the cerebral microbleed study groups and their contributions to this field of study in CMBs [1,2,116,117,118,119,120,121,122]. 

## 7. Future Directions

Late-onset Alzheimer’s disease (LOAD) is responsible for 95% versus genetic or familial early-onset AD (EOAD) for 5% of the extracellular neuritic Aβ plaques and intracellular hyperphosphorylated neurofibrils that form neurofibrillary tangles and contribute to neurodegeneration (ND). CAA amyloid (Aβ1–40) occurs primarily in the VSMC media and adventitial layers of small arteries, arterioles, and leptomeninges regions and results in vascular dysfunction due to loss of VSMC (apoptosis) and thinning of the media with microvessel dysfunction, loss of integrity with leakage of luminal blood contents due to extravasation via rhexis or rupture and/or diapedesis. Additionally, CAA is an age-related disease that contributes to neurodegeneration. VaD is caused primarily by HTN with associated SVD that includes CMBs, arteriolosclerosis or hyalinosis/lipohyalinosis, and even intracerebral atherosclerosis as a microvascular hypertensive vasculopathy with damage to the VSMC media with extravasation due to loss of VSMC with thinning and leakage of luminal blood contents due to extravasation via rhexis or rupture and or leakage. There exists considerable variability in the estimated prevalence of VaD. However, Kling et al. suggest that its prevalence is between 11–18% [123]. Jellinger states that its prevalence is between 8–10% [124]. Each of these three (LOAD, CAA, and VaD) contributes to clinical risk for the development of neurodegeneration and also share a common factor of being age-related [125].

Since LOAD, CAA, and VaD share many overlaps, intersects, and co-exist in age-related neurodegeneration (ND) and associate with CMBs, should we begin thinking about ND with a new perspective or begin thinking about new paradigm shifts in regard to the role of vascular diseases and neurodegeneration? Kling et al. have recently discussed this in greater depth and provided some suggestions [123]. There is considerable overlap in risk factors and findings at autopsies that are found in LOAD, CAA, and VaD (Alzheimer’s disease and Vascular dementia) [69,123]. As a result of the above discussions, we should consider a paradigm shift in which the focus of research is shifted to also include the multiple shared and overlapping risk factors, autopsy findings, and the associated interacting mechanisms [69,123,124,125,126]. Maybe we should be thinking more about utilizing the terms mixed dementias (LOAD and VaD) and co-occurring dementias more often [67].

In summary, this narrative review has strongly suggested, outlined, and discussed in Section 2.1 and Section 2.2. that BEC*act/dys* and BBB***dd*** play important roles in the development of CMBs and strokes (either ischemic or hemorrhagic). However, the current discussion regarding these important roles may need further confirmation in larger longitudinal studies in human individuals by MRI with stronger magnification strength, such as 7Tesla, to better identify CMB numbers that are occurring in either lobar or deep BG—infratentorial regions. 

## Figures and Tables

**Figure 1 biomedicines-12-01463-f001:**
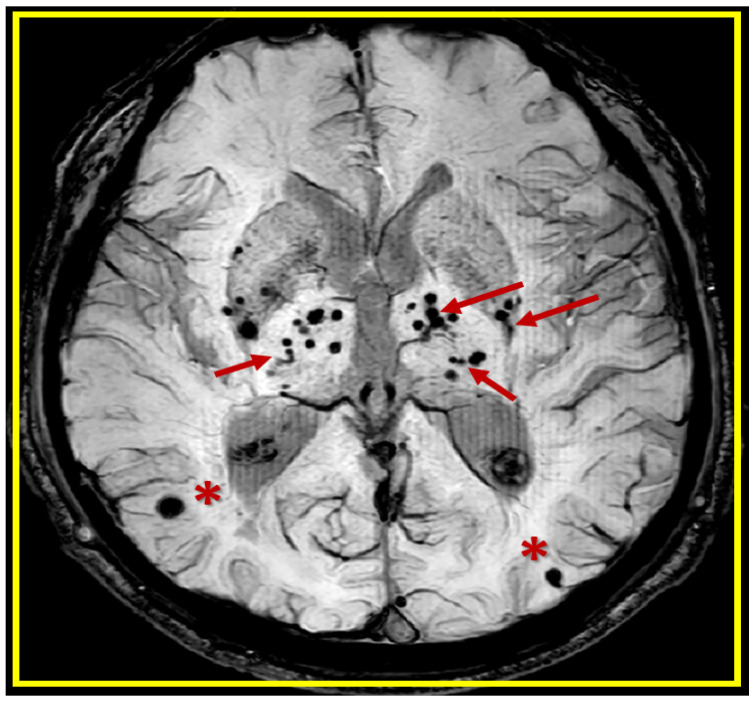
Cerebral microbleeds (CMBs) pathologic lesions with varying sizes (usually 2–5 mm and less than 10 mm) and different locations (lobar (red asterisks)) versus deep, infratentorial, white matter basal ganglia (BG) (red arrows) in T*2-weighted gradient recall echo (GRE)/susceptibility-weighted images (SWI) MRI images. This revised image was provided with permission by CC 4.0 [4].

**Figure 2 biomedicines-12-01463-f002:**
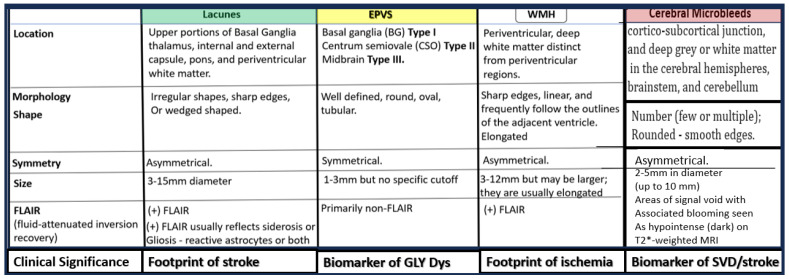
Comparing similarities and differences between the five identifiable structures of cerebral small vessel disease (SVD) in a table-like Figure. 1. Lacunes (a footprint of stroke); 2. enlarged perivascular spaces (EPVS) (a biomarker of glymphatic (GLY) system pathway dysfunction (dys)); 3. white matter hyperintensities (WMH) (footprint of ischemia); 4. cerebral microbleeds (CMBs) (biomarkers of SVD/stroke with hemorrhage or ischemic infarct); 5. recent small subcortical infarcts (historical or MRI findings of recent infarction similar to lacune parameters but with greater flair suggesting recent occurrence, not presented in this table-like Figure). The location of CMBs has further clinical importance in that lobar/cortical CMBs are CAA-related and deep, basal, infratentorial CMBs are hypertension-related. Revised table-like Figure image provided with permission by CC 4.0 [13]. CAA, cerebral amyloid angiopathy; mm, millimeter.

**Figure 3 biomedicines-12-01463-f003:**
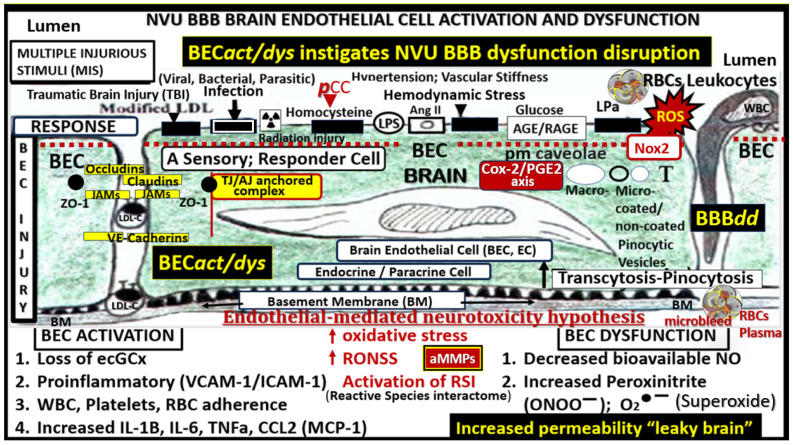
Brain endothelial cell activation and dysfunction (BEC*act/dys*) instigates neurovascular unit blood–brain barrier dysfunction/disruption (NVU BBB***dd***) and is responsible for the development of markers for small vessel disease (SVD), including cerebral microbleeds (CMBs). Initially, note the red-dashed line at the top of this image since it demarks the location of the multiple injurious species that are responsible for the initial brain endothelial cell injury in multiple clinical diseases and structural abnormalities, which importantly included SVD and CMBs. Also, note that below the BEC abluminal surface the multiple molecular consequences of the luminal multiple injurious stimuli that may be directly or indirectly related to the development of CMBs as well as other structural changes of SVD found on MRI. Please note that pathogen-associated molecular patterns (PAMPs) and damage-associated molecular patterns (DAMPs) are not shown; however, peripheral cytokines and chemokines are related to these signaling molecules. BH4 uncoupling is not depicted. Note the white background red lettering box of Nox2 with its important role of generating BEC-derived reactive oxygen species (ROS) in addition to mitochondrial-derived ROS and red background white lettering box of Cox2/PGE2 axis promoting a proinflammatory milieu with vasodilation and increased permeability**.** This highly modified image was provided with permission by CC 4.0 [16]. Ang II, angiotensin two; BBB, blood–brain barrier; BEC, brain endothelial cell; BECact/dys, brain endothelial cell activation/dysfunction; BH4, tetrahydrobiopterin; CCL2, chemokine (C-C motif) ligand 2; Cox-2, cyclo-oxygenase-2; Cox-2/PGE2 axis, cyclo-oxygenase-2; Prostaglandin E2; ecGCx, endothelial glycocalyx; intercellular adhesion molecule-1; ICAM-1; IL-1β; interleukin-1β; IL-6; interleukin-6; JAMs, junctional adhesion molecules; LDL, low density lipoprotein cholesterol; LPa, lipoprotein little a; MCP-1, monocyte chemotactic protein-1; NO, nitric oxide; Nox2, (NADPH Ox (nicotinamide adenine dinucleotide phosphate oxidase); ONOO, peroxinitrite; ***pns***CC, peripheral nervous system cytokines and chemokines; peroxinitrite; NVU, neurovascular unit; peroxinitrite; RBC, red blood cell; RONSS, reactive oxygen, nitrogen, sulfur species; ROS, reactive oxygen species; RSI, reactive species interactome; T, transcytosis; TNFα, tumor necrosis factor alpha; VCAM-1, vascular cellular adhesion molecule-1; WBC, white blood cell.

**Figure 4 biomedicines-12-01463-f004:**
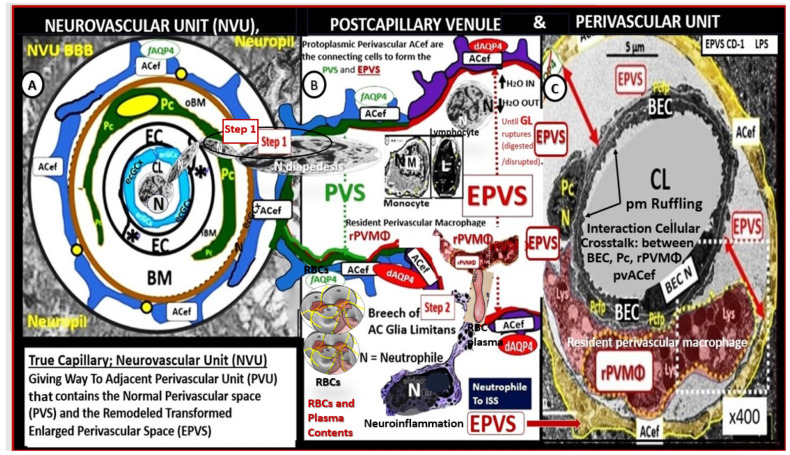
Comparison of the true capillary neurovascular unit (NVU) to the postcapillary venule perivascular unit (PVU). The NVU protoplasmic perivascular astrocyte endfeet (pvACef) (pseudo-colored blue) within the true capillary illustration. (**A**) Hand-drawn and pseudo-colored control true capillary neurovascular unit (NVU). Perivascular astrocyte end feet (PcACef-ACef) are the connecting and creating cells that allow remodeling of the normal perivascular unit (PVU) (**B**) perivascular spaces (PVS) that are capable of transforming and remodeling into the pathologic enlarged perivascular space (EPVS), which measure 1–3 mm on magnetic resonance imaging. Note that when the brain endothelial cells (BECs) become activated and NVU blood–brain barrier (BBB) disruption develops due to BEC activation and dysfunction (BECact/dys) (from multiple causes), there develops an increased permeability of fluids, peripheral cytokines and chemokines, and peripheral immune cells with a neutrophile (N) depicted herein penetrating the tight and adherens junctions (TJ/AJs) paracellular spaces to enter the postcapillary venule along with monocytes (M) and lymphocytes (L) into the postcapillary venule PVS of the perivascular unit (PVU). Note in panel (**B**) that this image illustrates step one of the two-step process of neuroinflammation. The postcapillary venule contains the PVU, which includes the normal PVS that has the capability to remodel the pathological EPVS. Also, note how the proinflammatory leukocytes enter the PVS along with fluids, solutes, and peripheral and endothelial cell-derived cytokines/chemokines from an activated, disrupted, and leaky NVU in panel (**A**). Importantly, the pvACef (pseudo-colored blue) and its glia limitans (pseudo-colored brown in the control NVU in (**A**) to the cyan color with exaggerated thickness for illustrative purposes in (**B**) that faces and adheres to the NVU BM extracellular matrix and faces the PVS PVU lumen. Also, note how the glia limitans becomes pseudo-colored red once the EPVS have developed and then becomes breeched due to activation of matrix metalloproteinases and degradation of the proteins in the glia limitans, which allow neurotoxins and proinflammatory cells to leak into the interstitial spaces of the neuropil and mix with the ISF and result in neuroinflammation (step two) of the two-step process of neuroinflammation [36]. Panel (**C**) depicts the aberrant EPVS with an aberrant, reactive perivascular macrophage (rPVMΦ; pseudo-colored red) from the lipopolysaccharide (LPS) treated model. Scale bar = 5 μm. This highly modified image is provided with permission by 4.0 [15]. ACef, perivascular astrocyte endfeet; AQP4, aquaporin 4 water channel.; Asterisk, tight and adherens junction; BBB, blood–brain barrier; BM, both inner (i) and outer (o) basement membrane; dACef and dpvACef, dysfunctional astrocyte endfeet; EC, brain endothelial cell; ecGCx, endothelial glycocalyx; EVPS, enlarged perivascular space; fAQP4, functional aquaporin 4; GL, glia limitans; H_2_O, water; L, lymphocyte; M, monocyte; N, neutrophile, and neuron; Pc, pericyte; PVS, perivascular space; L, lymphocyte; M, monocyte; PVU, perivascular unit; RBC(s), red blood cells; rPVMΦ, resident perivascular macrophage; TJ/AJ. Tight and adherens junctions.

**Figure 5 biomedicines-12-01463-f005:**
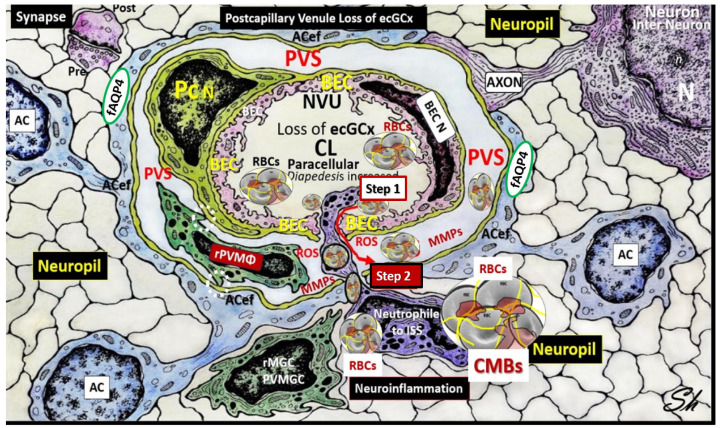
Perivascular astrocyte endfeet (ACef), neurovascular unit (NVU), perivascular unit (PVU), perivascular space PVS, and enlarged perivascular space (EPVS). The NVU is located centrally; note the absence of the endothelial glycocalyx (ecGCx) surface layer, which occurs in many neurovascular and neurodegenerative diseases with impaired cognition that also include obesity, metabolic syndrome (MetS), and type 2 diabetes mellitus (T2DM). Increased NVU permeability via BEC*act/dys* and blood–brain barrier (BBB) dysfunction/disruption (BBB***dd***) due to multiple clinical neurovascular and neurodegenerative diseases, which allows the entry of proinflammatory leukocytes into the PVU PVS in postcapillary venules. The accumulation of proinflammatory cells and oxidative stress with increased ROS will activate local and regional matrix metalloproteinases (MMPs)—proteolytic enzymes capable of degrading the glia limitans of the pvACef to allow the breeching of the postcapillary perivascular space and the entry of proinflammatory leukocytes, red blood cells (RBCs), solutes, and neurotoxins into the interstitial spaces (ISSs) to result in cerebral microbleeds (CMBs), neuroinflammation and increased central nervous system cytokines and chemokines (***cns***C/C), impaired cognition, and neurodegeneration via synaptic and neuronal loss with neural atrophy. Note the isolated synapse (uncradled) in the upper left-hand side of the illustration. Image reproduced with permission by CC by 4.0 [37]. AC, astrocyte; ACef, perivascular astrocyte endfeet; fAQP4, functional aquaporin-4; BEC, brain endothelial cell; N, nucleus; n, = nucleolus; Pc, pericyte; PVU, perivascular unit; pvMGC, perivascular microglia cell; rMGC, reactive microglia cell; rPVMΦ, resident perivascular macrophage; ROS, reactive oxygen species. Modified image provided with permission by CC 4.0 [36,37].

**Figure 6 biomedicines-12-01463-f006:**
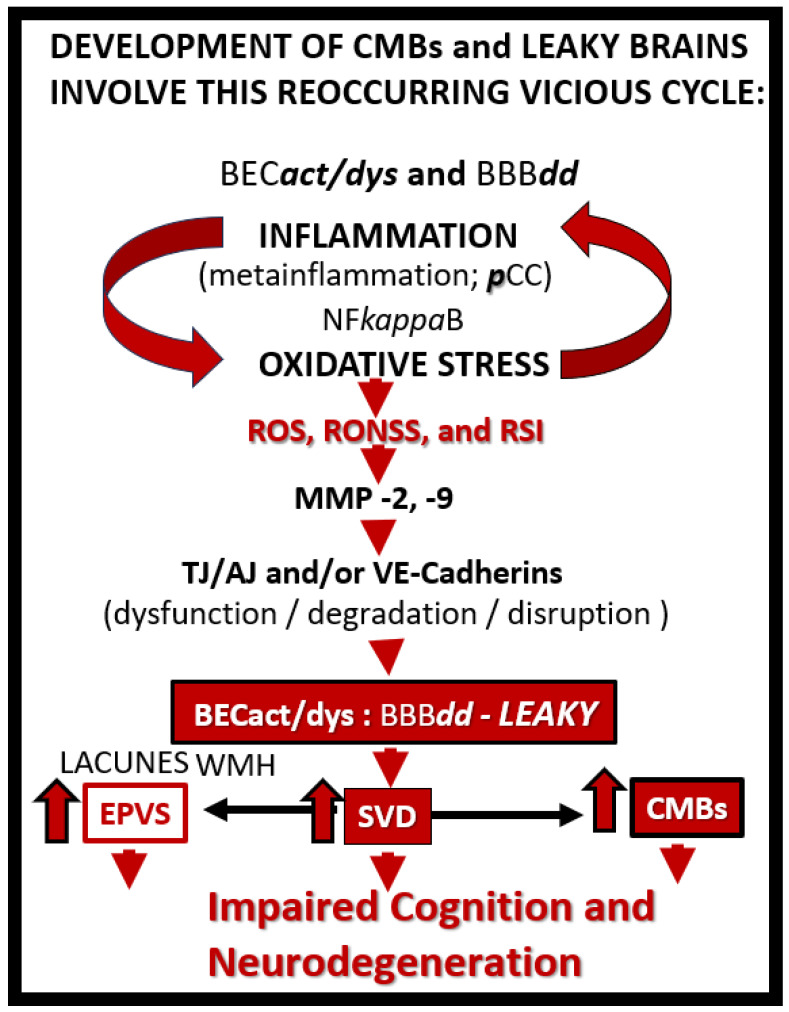
Possible reoccurring sequence of events resulting in a vicious cycle with increased cerebral microbleeds and leaky brains that lead to impaired cognition and neurodegeneration. Indeed, oxidative stress (reactive oxygen, nitrogen, and sulfur species referred to as the reactive species interactome (RSI)) and inflammation (including the injurious species of peripheral cytokines/chemokines (***p***CC) and the central nervous system cytokines/chemokines (***cns***CC)) are difficult to separate since they may act as a perpetual vicious cycle and each is individually capable of triggering the neurodegeneration cascade with neural or neural synapse dysfunction and or loss with subsequent impaired cognition in multiple neurologic diseases including SVD and specifically CMBs. Arrowheads, lead to; BECact/dys, brain endothelial cell activation/dysfunction; BBB***dd*,** blood–brain barrier dysfunction, disruption; CMBs, cerebral microbleeds; EPVS, enlarged perivascular spaces; MMP, matrix metalloproteinases; NF*kappa*B, nuclear factor kappa B; ***p***CC, peripheral cytokines/chemokines; ROS, reactive oxygen species; RONSS, reactive oxygen, nitrogen, sulfur, species; RSI, reactive species interactome; SVD, small vessel disease; TJ/AJ, tight and adherens junctions; VE-Cadherins, vascular endothelial cadherins; WMH, white matter hyperintensities.

**Figure 7 biomedicines-12-01463-f007:**
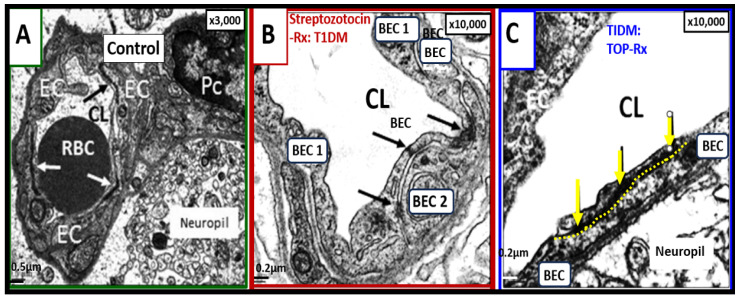
Attenuation and/or loss of tight and adherence junction(s) (TJ/AJs) paracellular blood–brain barrier (BBB) in male CD-1 streptozotocin-induced (STZ) diabetic preclinical mice models resulting in blood–brain barrier dysfunction and disruption (BBB***dd***) protected by the carbonic anhydrase inhibitor (Topiramate a mitochondria antioxidant) in the mid brain as compared to the cerebellum. The STZ-induced type I diabetic mice revealed disruption of the BBB by ^14^C-sucrose measurements. Panel (**A**) demonstrates three prominent elongated electron-dense TJ/AJ (white and black arrows). Panel (**B**) depicts a discontinuous and disrupted TJ/AJ (black arrows) into three distinct segments in the midbrain. Note how the TJ/AJ tends to form at EC-EC overlap junctions. Panel (**C**) demonstrates that treatment with Topiramate prevented this disruption in the brain endothelial cell BBB (yellow and black arrows and yellow dashed line below the intact BBB TJ/AJ) in the midbrain. Revised figure images provided with permission by CC 4.0 [56]. Magnification ×3000; scale bar = 0.5 μm (**A**); ×10,000; scale bar = 0.2 μm in (**B**,**C**).

**Figure 8 biomedicines-12-01463-f008:**
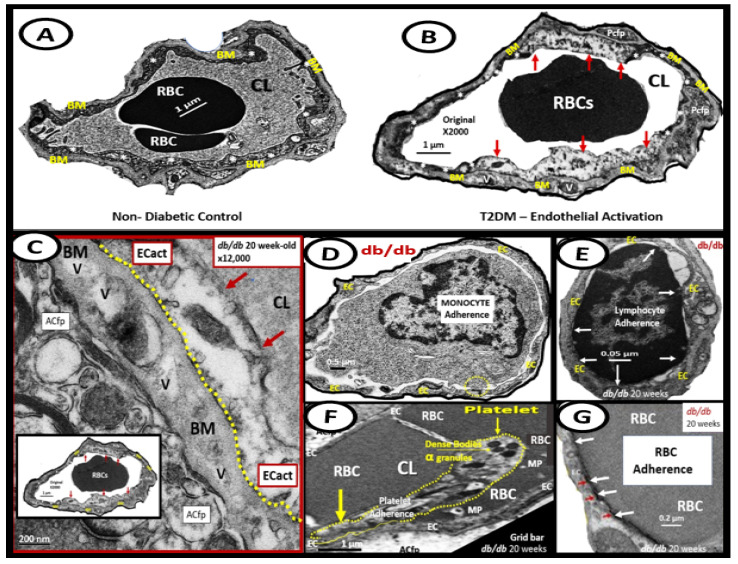
Ultrastructural images of brain endothelial cell activation/dysfunction (BEC*act/dys*) in 20-week-old obese diabetic *db/db* female models. (**A**) demonstrates the normal neurovascular unit (NVU) blood–brain barrier (BBB) capillary and notes the thinness of the moderate electron-dense cytoplasm. (**B**) Depicts regions of abrupt thickened electron-lucent (red arrows) with vacuolization of the basement membrane (BM) in obese diabetic female *db/db* models with BEC*act/dys as* compared to control panel A. (**C**) Depicts BM thickening with increased vacuole-like bodies (V). (**D**–**G**) Depict monocyte (**D**), lymphocyte (**E**) leukocytes, platelet (**F**), and red blood cell (**G**) adherence to the plasma membrane of brain endothelial cells in BEC*act/dys db/db* models. Images reproduced with permission by CC 4.0 [13,83]. Original magnification = ×2000; scale bar = 1 μm. ACfp, astrocyte foot processes; Cl, capillary lumen; EC, brain endothelial cells; MP, microparticle of the platelet.

**Figure 9 biomedicines-12-01463-f009:**
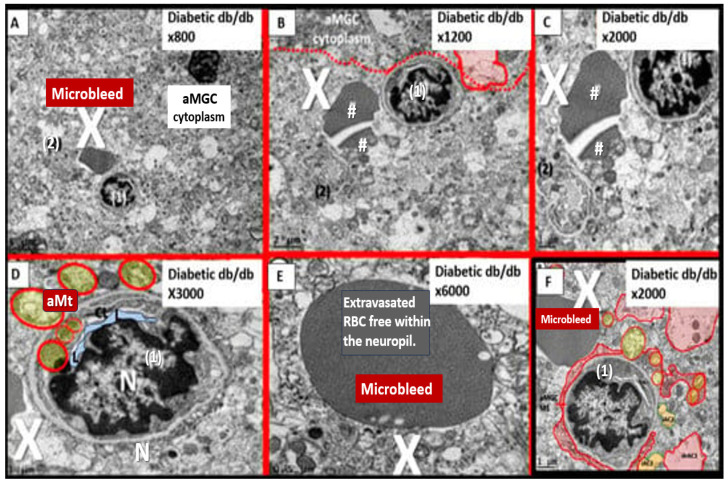
Six-panel image depicting cerebral microbleeds-hemorrhages in preclinical female obese metabolic syndrome and type 2 diabetes mellitus *db/db* genetic mice models from frontal cortical layer III. Each of these six panels, except for panel (**D**), depicts a cerebral microbleed identified by a large white X. Note in panels (**B**,**C**), the homogeneous electron-dense staining could also represent free extruded plasma. Also, it is important to note that in panel (**E**), the homogenous electron-dense staining may represent a free RBC within the neuropil due to the presence of the multiple small electron lucent vacuoles (white in color) that aid in the identification of this being an RBC. Note in panel (**D**) the aberrant mitochondria (pseudo-colored yellow outlined in red) in the brain endothelial cell and also in the adjacent aMGC, which suggest activation of brain endothelial cells. Panels (**A**) and (**F**) depict cerebral microbleeds and low magnification and high magnification, respectively. Images provided by CC 4.0 [15,83]. aMGC, reactive microglia cell; hashtags, microbleeds most likely plasma; N, nucleus; numbers (1), (2), adjacent capillary neurovascular units; RBC, red blood cell; X, microbleeds. Magnifications and scale bars vary from image to image and are present in each panel.

**Figure 10 biomedicines-12-01463-f010:**
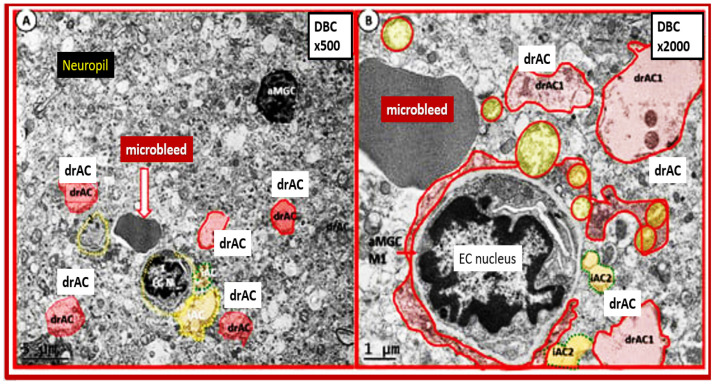
Extrusion of a free RBC into the neutrophil from an adjacent capillary neurovascular unit in the cortical layer III from the diabetic *db/db* model at low and high magnification. Note the labeled microbleed associated with the nearby small NVU capillary, the encroaching reactive microglia cell M1-like proinflammatory cell (aMGC M1—like) cytoplasmic process enveloping the aberrant small neurovascular unit capillary, and the detached and retracted astrocytes pseudo-colored red (drAC1) and a couple of intact attached astrocytes (pseudo-colored yellow) iAC2. Images provided by CC 4.0 [83] Magnification x800; scale bar = 1 μm (panel (**A**)) and Magnification x2000; scale bar = 1 μm (panel (**B**)). EC N, brain endothelial cell nucleus; aMGC, reactive microglia cell.

**Figure 11 biomedicines-12-01463-f011:**
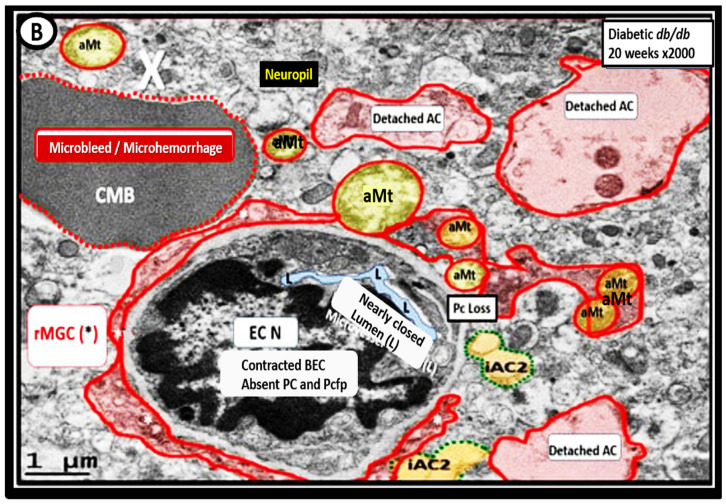
A microbleed plasma extrusion (~5 μm) immediately adjacent to a contracted lumen microvessel (~5 μm). Note how the lumen of this microvessel (pseudo-colored light blue) is nearly collapsed and that the brain endothelial cell (BEC) nucleus is contracted with extremely prominent chromatin condensation instead of being heterogenous, suggesting BEC activation and dysfunction. These similar morphological contracted BEC remodeling changes and nuclear remodeling changes were observed in the aortic endothelium of activated endothelial cells in female Western diet-fed mice at 20 weeks of age. Also, note that the reactive microglia (pseudo-colored red) encircle this microvessel that it contains multiple aberrant mitochondria (aMt), which provide excessive mitochondria-derived reactive oxygen species that provide BEC injury for the response to injury wound healing mechanisms at the level of this microvessel to result in BEC activation and dysfunction. Importantly, note reactive astrocyte detachment and separation of reactive perivascular astrocytes. These remodeling changes allow for microvessel disruption and microbleeds. Image provided by CC 4.0 [83,84]. AC, astrocyte; asterisk, reactive microglia; CMB, cerebral microbleed; EC N, brain endothelial nucleus; iAC, intact attached astrocyte; rMGC, reactive microglia cell; Pc, pericyte; X, microbleed-microhemorrhage.

**Figure 12 biomedicines-12-01463-f012:**
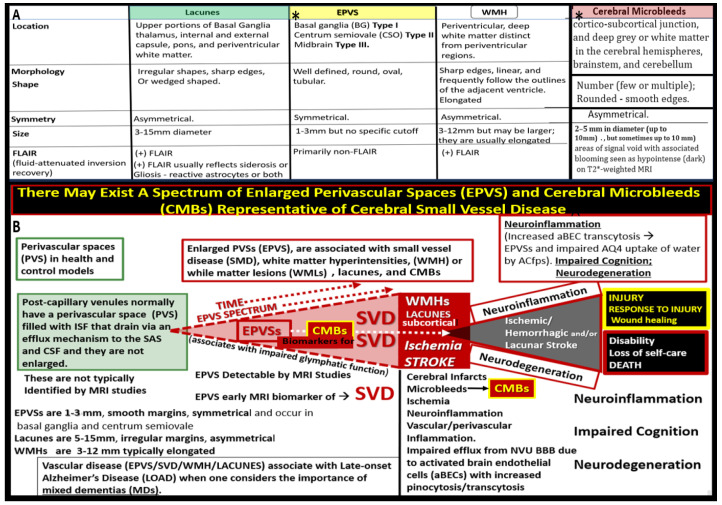
Small vessel disease (SVD) comparisons and EPVS and CMBs spectrum disorders. Upper panel A depicts the comparisons between the four major components of SVD: lacunes, enlarged perivascular spaces (EPVS), white matter hyperintensities (WMH), and cerebral microbleeds (CMBs). Note that panel A is a reproduction of the previous Figure 2. Lower panel B depicts the spectrum of SVD with emphasis on EPVS and CMBs. Note: regarding the importance of EPVS and CMBs, each is now considered to be biomarkers for microvessel SVD, and that time plays an important role in the de EPVS and CMBs that are 2 of the 4 components of SVD biomarkers. aBEC, activated brain endothelial cells; ACfps, astrocyte foot processes; AQ4, aquaporin 4; Asterisk, denotes the importance of enlarged perivascular spaces (EPVS); CSF, cerebral spinal fluid; ISF, interstitial fluid; LOAD, late-onset Alzheimer’s disease; MRI, magnetic resonance image or imaging; PVS. perivascular spaces; SAS, subarachnoid space.

**Figure 13 biomedicines-12-01463-f013:**
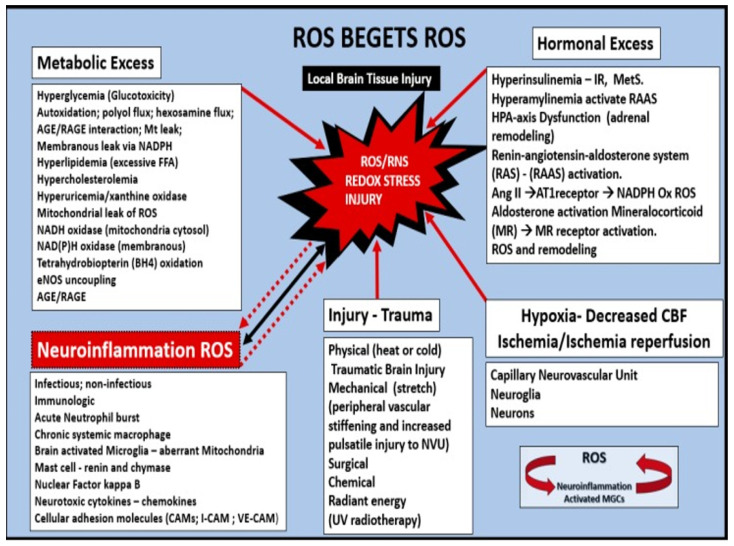
Multiple inputs (peripheral and central nervous system (CNS)) for ROS generation in the brain including the importance of ROS begetting ROS. This Figure illustrates how the metabolic and hormonal excesses of the MetS interact to produce reactive oxygen/nitrogen species (ROS/RNS) and redox stress. Note how neuroinflammation, brain injury hypoxia-ischemia, and reperfusion may all work individually or synergistically to produce redox stress damage to the brain, which results in accelerated aging, and neurodegeneration and thus supports the oxidative-redox stress hypothesis to LOAD, CAA, and CMBs. It seems that at every turn of brain injury with its response to injury wound healing event the Oxidative—redox stress/MMP activation axis comes into play. This places the activation of the ROS/MMP axis at the very center and results in it becoming a critical player in both abnormal structural and functional changes, which ultimately results in neurodegeneration and impaired cognition. Image reproduced with permission by CC 4.0 [69]. AGE/RAGE, advanced glycation end products/receptor for advanced glycation end products; Ang II, angiotensin II; AT1R, angiotensin type 1 receptor; eNOS, endothelial nitric oxide synthase; HPA, hypothalamic pituitary adrenal; MC, mast cell; MGCs, microglia cells; NADPH—NADPH Ox, reduced nicotinamide adenine dinucleotide phosphate oxidase; NVU, neurovascular unit; NF-kB, nuclear factor-kappa B; RAS, renin-angiotensin system; RAAS, renin angiotensin aldosterone system; ROS/RNS, reactive oxygen species/reactive nitrogen species.

**Figure 14 biomedicines-12-01463-f014:**
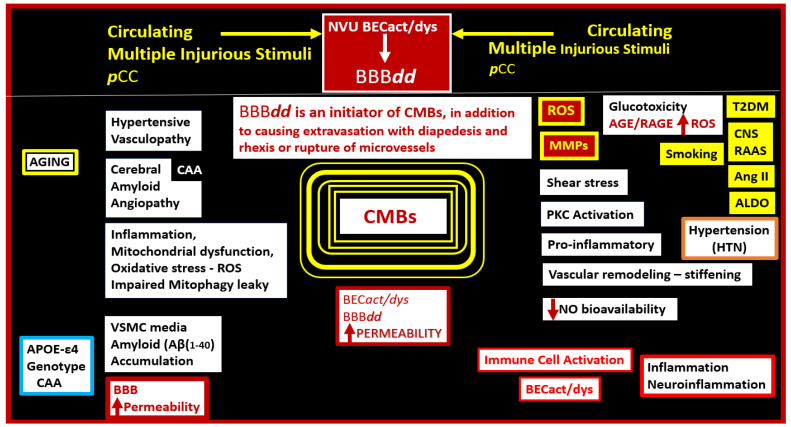
Schematic for aging, APOE-ε4 genotype, T2DM, HTN, and inflammation/neuroinflammation in the development of cerebral microbleeds (CMBs). Note that each of the major color-coded components AGING yellow, APOE-ε4 (blue), Hypertension (orange), Inflammation/Neuroinflammation (red), and their respective components may all contribute to the development and evolution of the centrally located CMBs. Also, note the importance of BEC*act/dys* and BBB***dd*** as a possible final common pathway for the development of CMBs. Note that ROS and MMPs are placed in a colored dark-red box with yellow lettering since they are possibly the final common signaling pathway for the development of CMBs due to the degradation of the glia limitans of the perivascular units’ perivascular spaces outermost limiting barrier membrane and the rupture of arterioles in CAA. APOE-ε4, apolipoprotein epsilon 4; BBB, blood–brain barrier; BBB***dd***, blood–brain barrier dysfunction and/or disruption; BEC*act/dys*, brain endothelial cell activation and dysfunction; CAA, cerebral amyloid angiopathy; MMPs, matrix metalloproteinases; NO, nitric oxide; PKC, protein kinase C; ROS, reactive oxygen species, upward arrow, increase; downward arrow, decrease.

**Figure 15 biomedicines-12-01463-f015:**
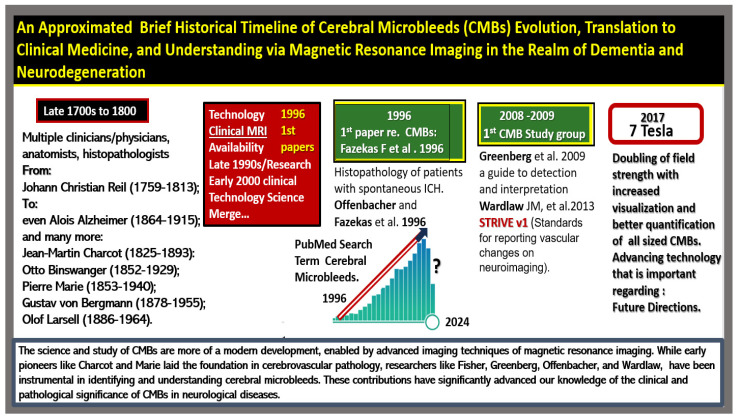
Historical Timeline of Cerebral Microbleeds (CMBs). Cerebral small vessel disease (SVD) is the most common vascular disease that affects the whole brain in its entirety. CMBs are one of the prominent findings amongst SVD findings that became readily apparent once MRI had developed to demonstrate paramagnetic properties, and they are associated with not only being a marker of SVD but also having an association with an increased risk of larger ICH associated with clinical neurologic deficits in addition to ischemic strokes that are symptomatic. SVDs are known to develop gradually with white matter hyperintensities and lacunes being evaluated earlier (1980s and early 1990s) and CMBs (late 1990s–to date) and more recently perivascular spaces (2023–2024).

**Table 1 biomedicines-12-01463-t001:** Similarities and differences between cerebral amyloid angiopathy (CAA) and hypertension (HTN) as related to the formation of cerebral microbleeds (CMBs). Aβ (1–40), amyloid beta 1–40; ALDO, aldosterone; Ang II, angiotensin two; ApoE-ε4, apolipoprotein E epsilon 4 genotype; BBB***dd***, blood–brain barrier dysfunction and/or disruption; BEC(s), brain endothelial cells; BEC*act/dys*, brain endothelial cell activation and dysfunction; CNS, central nervous system; ECM, extracellular matrix; RAAS, brain renin-angiotensin-aldosterone system; SNS, sympathetic nervous system; VSMCs, vascular smooth muscle cells.

Factor	Cerebral Amyloid Arteriopathy (CAA)	Hypertension (HTN)
Definition	CAA is a condition where primarily amyloid beta Aβ (1–40) protein builds up in the walls of arteries and arterioles, primarily to the media VSMCs, ECM, and adventitia in the CNS, increasing the risk of cerebral microbleeds (CMBs).	Hypertension refers to high blood pressure, which can lead to changes in small blood vessels (microvessels) in the brain, potentially causing microbleeds (CMBs).
Molecular	Cerebral amyloid beta (Aβ1–40) accumulation in microvessels resulting in loss of integrity to the vessel wall with extrusion of blood contents by rhexis (rupture) and/or diapedesis.	Peripheral and CNS RAAS activation, SNS, increased AngII and ALDO. Multifactorial genetic and environmental causes lead to dysfunction and damage to intimal BECs, media VSMCs with loss of integrity, and extrusion of luminal blood contents by rhexis (rupture) and/or diapedesis.
Location	Lobar (cortex, gray–white matter junction, subcortical white matter, and leptomeninges) location.	Deep (infratentorial, basal ganglia, lacunal, internal and external capsule, thalamus, and brainstem).
Mechanisms—Pathogenesis	Vascular rupture due to endothelial and VSMC remodeling with loss of vascular integrity and rupture due to Aβ (1–40) direct effect on VSMC and ECM. BEC*act/dys* and BBB*dd.*	Vascular stiffness, increased pulse pressure, vascular hyalinosis, atherosclerosis, arteriolosclerosis, oxidative stress, and inflammation. BEC*act/dys* and BBB*dd*.
Associated conditions	Often seen in conjunction with Alzheimer’s disease and older age ≥ 60.	Age of onset may vary but usually increases with aging and can lead to various cardiovascular conditions such as heart disease and stroke.
Risk factors	Aging, genetics (e.g., presence of ApoE-ε4 allele genotype), and history of Alzheimer’s disease	Obesity, sedentary lifestyle, smoking, excessive alcohol consumption, and family history of hypertension.

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
