# Peer review of "Cerebral Microbleeds Associate with Brain Endothelial Cell Activation-Dysfunction and Blood–Brain Barrier Dysfunction/Disruption with Increased Risk of Hemorrhagic and Ischemic Stroke"

_biomedicines, 2024, doi:10.3390/biomedicines12071463_

Round 1
Reviewer 1 Report
Comments and Suggestions for Authors
Review for the manuscript: Cerebral Microbleeds Associate with Brain Endothelial Cell Activation-Dysfunction and Blood-Brain Barrier Dysfunction- Disruption with an Increased Risk of Hemorrhagic and Is- chemic Stroke That Is Capable of Exacerbating Neurodegeneration
Dear Doctor Masaru Tanaka, Guest Editor,
Thank you for the invitation to review this manuscript submitted to Biomedicines. After reading it, I have some suggestions before it can be accepted for publication.
OVERALL COMMENTS
To summarize, this is an exciting narrative review focusing on BECact/dys and BBBdd, which can play important roles in developing Cerebral Microbleeds and strokes.
TITLE
The title is not adequate. It is too long, and it is not clear what kind of article this is. From the title, I thought it was an original article. I suggest reducing the length of the title to include that it is a review article.
ABSTRACT
The abstract is fine; however, in line 20 where we can read, “CMBs (a global problem) are known to occur …”. I suggest that this sentence be included in the beginning:
“Cerebral microbleed(s) (CMBs) are increasingly and have been considered a global problem. This condition has been viewed not only as a marker for cerebral small vessel disease (SVD) but also as having an increased risk for the development of stroke (hemorrhagic/ischemic) and aging-related dementia….”
I suggest that the authors include this article in this section as a narrative review. Moreover, clarify the aim/objective of the review.
KEYWORDS
Please do not use MRI; NVU; expand the abbreviations.
INTRODUCTION and TEXT
I also suggest the inclusion of newer references. In PUBMED, it is possible to find several articles that were published in 2023 and in 2024.
The figures are nice. However, there is too much information in Figures 3, 4, 9,10, 11, and 12. I suggest re-elaborating them. The main concern regards Figures 3, 4, 9, 10, and 12.
Figures 5 to 11 are not of very good quality. It is not possible to read some words.
Figure 13 is fine.
Please check the font and size. It seems that they vary along with the text. The same for space-lining.
The space between the lines in Table 1 can be reduced. Please check the font.
CONCLUSION
This section is adequate; however, I suggest the authors also include the limitations of this review.
Author Response
Open Review
Comments and Suggestions for Authors
OVERALL COMMENTS
To summarize, this is an exciting narrative review focusing on BECact/dys and BBBdd, which can play important roles in developing Cerebral Microbleeds and strokes.
Response to reviewers #1: Author wishes to thank reviewer #1 for this kind and generous comment.
TITLE
The title is not adequate. It is too long, and it is not clear what kind of article this is. From the title, I thought it was an original article. I suggest reducing the length of the title to include that it is a review article.
Response to reviewer #1: Author agrees and has shortened the title in blue letter removing “Capable of Exacerbating Neurodegeneration” lines 1 to 4 and added that it is review upper left of title by Journal.
ABSTRACT
The abstract is fine; however, in line 20 where we can read, “CMBs (a global problem) are known to occur …”. I suggest that this sentence be included in the beginning:
Response to reviewer #1: Author has now placed this global problem in the beginning as follows: “Globally, cerebral microbleeds (CMBs) are increasingly being viewed… line (12).
I suggest that the authors include this article in this section as a review (done). Moreover, clarify the aim/objective of the review.
Response to reviewer #1: Review now appears just prior to the beginning of the title so all will know it is a review. The aim/objective of the review now appears within the introduction (per recommendation of the academic editor and see that response to academic editor as follows: The primary objective of this narrative review is to provide the reader with an increased database and understanding of CMBs, their evolution via BECact/dys and BBBdd with increased permeability and increased risk of hemorrhagic and ischemic stroke that are capable of exacerbating neurodegeneration since abstract already has over 200 word count. In lines (129-132).
KEYWORDS
Please do not use MRI; NVU; expand the abbreviations.
Response to reviewer #1: This has been corrected as follows: Keywords: Alzheimer’s disease; Blood-brain barrier; Cerebral amyloid angiopathy; Cerebral microbleeds; Dementia; Hypertension; Magnetic resonance imaging; Neurovascular unit; Cerebral small vessel disease; Transmission electron microscopy in lines (28-31).
INTRODUCTION and TEXT
I also suggest the inclusion of newer references. In PUBMED, it is possible to find several articles that were published in 2023 and in 2024.
Response to reviewer #1 as follows: Academic editor has suggested that author expand references and many of these new references are from manuscripts published In 2003-2004.
The figures are nice.
Author wishes to thank reviewer #1 for this kind comment.
However, there is too much information in Figures 3, 4, 9,10, 11, and 12. I suggest re-elaborating them. The main concern regards Figures 3, 4, 9, 12
Response to reviewer #1: Authors agree that some figures have too much information imbedded within the figures. However, since this narrative has the primary purpose for advancing knowledge in CMB, it is thought that the readers will be able to work with this excess information and allow for increased knowledge extraction from these figures.
Figures 5 to 11 are not of very good quality. It is not possible to read some words.
Response to reviewer #1: Author has gone back and improved these figures to improve their quality and especially improved figures 9, 10, 11 TEMs of CBM in the diabetic db/db mouse models at 20-weeks of age.so that labeling is now improved in clarity.
Figure 13 is fine.
Please check the font and size. It seems that they vary along with the text. The same for space-lining.
Response to reviewer #1: In most instances the font size is varied such that it fits the image, illustration, and/or schematic within the limited space size of the entire image. Author has chosen to allow the varying font size and spacing.
The space between the lines in Table 1 can be reduced. Please check the font.
Response to reviewer #1: Author agrees and has been able to decrease them some.
CONCLUSION
This section is adequate; however, I suggest the authors also include the limitations of this review.
Response to reviewer #1: Author has now placed a comment regarding the papers strength and its limitations as follows: “While narrative reviews provide a flexible, yet rigorous platform to approach a topic of interest and present knowledge synthesis that is extremely useful for sharing knowledge and information with other educators and researchers, they also have inherent limitations. in that, the biases of the author(s) may readily creep into the review}. (lines 695-699).
Sincerely with gratitude,
Melvin R Hayden
submitting author.

Reviewer 2 Report
Comments and Suggestions for Authors
This comprehensive review provides an in-depth analysis of the pathophysiology of cerebral microbleeds (CMBs), highlighting their importance as markers of cerebral small vessel disease and their association with stroke and dementia. The author meticulously examine the potential mechanisms underlying the development of CMBs, including activation and dysfunction of brain endothelial cells, disruption of the blood-brain barrier, and enlarged perivascular spaces. The author highlights the prevalence of CMBs in the aging population and their impact on health burden. The review is well supported by electron micrographs that illustrate the pathologic events leading to endothelial damage and the development of CMBs.
There are a few minor suggestions to improve the readability of the text. In Figure 5, the labeling of aquaporin-4 is missing, although it is mentioned in the figure legend. Some of the figures (3-5) are very confusing and it is recommended to reduce the number of labeled elements. Abbreviations for the same terms are repeated in several sections, making the text confusing.
Overall, this review is a valuable resource for researchers and clinicians seeking a deeper understanding of CMBs and their impact on neurological health.
Author Response
Cover Letter: CMBs Reviewer 2 round 1 Biomedicines 6 10 24
Open Review
First the author would like to thank reviewer #2 for the precious time required to review this manuscript and the knowledge and recommendations made to improve this submitted manuscript.
This comprehensive review provides an in-depth analysis of the pathophysiology of cerebral microbleeds (CMBs), highlighting their importance as markers of cerebral small vessel disease and their association with stroke and dementia. The author meticulously examine the potential mechanisms underlying the development of CMBs, including activation and dysfunction of brain endothelial cells, disruption of the blood-brain barrier, and enlarged perivascular spaces. The author highlights the prevalence of CMBs in the aging population and their impact on health burden. The review is well supported by electron micrographs that illustrate the pathologic events leading to endothelial damage and the development of CMBs.
Authors Response: Author wishes to the thank the reviewer for these kind words of support.
There are a few minor suggestions to improve the readability of the text. In Figure 5, the labeling of aquaporin-4 is missing, although it is mentioned in the figure legend. Some of the figures (3-5) are very confusing and it is recommended to reduce the number of labeled elements. Abbreviations for the same terms are repeated in several sections, making the text confusing
Authors Response: Functional aquaporin4 (fAQP4) has now been added to figure 5 in cyan green color. Author agrees with reviewer that figures 3 and 4 are busy and over labeled. However, author feels that this excessive labeling is necessary in order for the readers to fully appreciate the images and gain further knowledge and this is important to the original aim of the paper to increase the database of knowledge for the readers. Therefore author has chosen to not decrease the excessive labeling of images.
Overall, this review is a valuable resource for researchers and clinicians seeking a deeper understanding of CMBs and their impact on neurological health.
Authors response: Author wishes to the thank the reviewer for these kind words of support.
Sincerely with gratitude,
Melvin R Hayden
submitting author

Reviewer 3 Report
Comments and Suggestions for Authors
A very interesting review article on the role of cerebral microbleeds (CMB). The article details the role of CMB - as a marker of cerebral small vessel disease (SVD), but also as a factor increasing the risk of developing stroke (hemorrhagic/ischemic) and dementia associated with aging.
The authors discussed the role of activation and dysfunction of brain endothelial cells and dysfunction and/or disruption of the blood-brain barrier in the development of SVD, and the development and evolution of CMB.
The authors presented in detail the role of various factors in the development, activation and pathological processes at the cellular level with a possible sequence of events in the development of bleeding from cerebral microvessels:
1/ Brain Endothelial Cell activation and dysfunction (BECact/dys), 2/ Blood-Brain Barrier dysfunction and/or disruption (BBBdd) with increased permeability, 3/ Hypertensive (HTN) vasculopathy, 4/ Cerebral Amyloid Angiopathy (CAA).
The next chapters concerned the results of Transmission Electron Microscopy (TEM) Imaging of Brain Endothelial Cell activation and dysfunction (BECact/dys), Blood-Brain Barrier dysfunction and disruption (BBBdd) With Cerebral Microbleeds (CMBs), and the role of apolipoprotein E atrial fibrillation in the occurrence of CMBs .
Comments: only one note - the figures are difficult to interpret; of course, this is related to a lot of information contained, but you should consider simplifying it and maybe breaking it down into more pieces, with fewer details, which may be more readable?
Author Response
Open Review
Comments and Suggestions for Authors
First the author would like to thank reviewer #3 for the precious time, effort, and knowledge required to review this manuscript. Also, the helpful recommendations made to improve this submitted manuscript
A very interesting review article on the role of cerebral microbleeds (CMB). The article details the role of CMB - as a marker of cerebral small vessel disease (SVD), but also as a factor increasing the risk of developing stroke (hemorrhagic/ischemic) and dementia associated with aging.
Author wishes to thank reviewer # 3 for these kind words.
The authors discussed the role of activation and dysfunction of brain endothelial cells and dysfunction and/or disruption of the blood-brain barrier in the development of SVD, and the development and evolution of CMB.
The authors presented in detail the role of various factors in the development, activation and pathological processes at the cellular level with a possible sequence of events in the development of bleeding from cerebral microvessels:
1/ Brain Endothelial Cell activation and dysfunction (BECact/dys), 2/ Blood-Brain Barrier dysfunction and/or disruption (BBBdd) with increased permeability, 3/ Hypertensive (HTN) vasculopathy, 4/ Cerebral Amyloid Angiopathy (CAA).
The next chapters concerned the results of Transmission Electron Microscopy (TEM) Imaging of Brain Endothelial Cell activation and dysfunction (BECact/dys), Blood-Brain Barrier dysfunction and disruption (BBBdd) With Cerebral Microbleeds (CMBs), and the role of apolipoprotein E atrial fibrillation in the occurrence of CMBs.
Author thanks reviewer #3 for the kind remarks and suggestions and comments to improve this narrative review submitted
Comments: only one note - the figures are difficult to interpret; of course, this is related to a lot of information contained, but you should consider simplifying it and maybe breaking it down into more pieces, with fewer details, which may be more readable?
Authors response: Author agrees with reviewer # 3 that figures 3 and 4 are busy and over labeled and has given this considerable thought. However, author feels that this excessive labeling is necessary in order for the readers to fully appreciate the images and gain the most knowledge from these images and therefore, I have not decreased the excessive labeling in this revised submission, so that the readers can obtain the maximum amount of knowledge to be gleaned from these images. Further, author has improved the quality of labeling in all images such that they are now easier to read by readers of the manuscript if published in Biomedicines.
Sincerely with gratitude,
Melvin R Hayden
submitting author
